# Variation of the Start Date of the Vegetation Growing Season (SOS) and Its Climatic Drivers in the Tibetan Plateau

**DOI:** 10.3390/plants13081065

**Published:** 2024-04-10

**Authors:** Hanya Tang, Yongke Li, Xizao Sun, Xuelin Zhou, Cheng Li, Lei Ma, Jinlian Liu, Ke Jiang, Zhi Ding, Shiwei Liu, Pujia Yu, Luyao Jia, Feng Zhang

**Affiliations:** 1Chongqing Jinfo Mountain Karst Ecosystem National Observation and Research Station, School of Geographical Sciences, Southwest University, Chongqing 400715, China; tanghanya2000@126.com (H.T.); july334477@email.swu.edu.cn (X.S.); ljl1980522@email.swu.edu.cn (J.L.); jiangke@email.swu.edu.cn (K.J.); dingzhi11@mails.ucas.ac.cn (Z.D.); liushiwei@swu.edu.cn (S.L.); yupujia@swu.edu.cn (P.Y.); haohao21@email.swu.edu.cn (L.J.); yh1234x@email.swu.edu.cn (F.Z.); 2College of Computer and Information Engineering, Xinjiang Agriculture University, Urumqi 830052, China; 3Zhuhai Orbita Aerospace Science & Technology Co., Ltd., Zhuhai 519080, China; zhouxuelin@live.cn; 4Observation and Research Station of Ecological Restoration for Chongqing Typical Mining Areas, Ministry of Natural Resources, Chongqing Institute of Geology and Mineral Resources, Chongqing 401120, China; licheng@cqdky.com (C.L.); malei@cqdy.com (L.M.); 5Wansheng Mining Area Ecological Environment Protection and Restoration of Chongqing Observation and Research Station, Ministry of Natural Resources, Chongqing Institute of Geology and Mineral Resources, Chongqing 400715, China

**Keywords:** Tibetan Plateau, start date of the growing season, climate change, PLSR

## Abstract

Climate change inevitably affects vegetation growth in the Tibetan Plateau (TP). Understanding the dynamics of vegetation phenology and the responses of vegetation phenology to climate change are crucial for evaluating the impacts of climate change on terrestrial ecosystems. Despite many relevant studies conducted in the past, there still remain research gaps concerning the dominant factors that induce changes in the start date of the vegetation growing season (SOS). In this study, the spatial and temporal variations of the SOS were investigated by using a long-term series of the Normalized Difference Vegetation Index (NDVI) spanning from 2001 to 2020, and the response of the SOS to climate change and the predominant climatic factors (air temperature, LST or precipitation) affecting the SOS were explored. The main findings were as follows: the annual mean SOS concentrated on 100 DOY–170 DOY (day of a year), with a delay from east to west. Although the SOS across the entire region exhibited an advancing trend at a rate of 0.261 days/year, there were notable differences in the advancement trends of SOS among different vegetation types. In contrast to the current advancing SOS, the trend of future SOS changes shows a delayed trend. For the impacts of climate change on the SOS, winter T_max_ (maximum temperature) played the dominant role in the temporal shifting of spring phenology across the TP, and its effect on SOS was negative, meaning that an increase in winter T_max_ led to an earlier SOS. Considering the different conditions required for the growth of various types of vegetation, the leading factor was different for the four vegetation types. This study contributes to the understanding of the mechanism of SOS variation in the TP.

## 1. Introduction

Vegetation phenology refers to the rhythmic growth and developmental stages closely associated with environmental changes throughout a plant’s life cycle [1,2,3]. It encompasses various processes such as germination, leafing, flowering, fruiting, and leaf shedding [4]. As a pivotal component of terrestrial ecosystems, vegetation serves as a natural link connecting elements such as soil, water, and the environment. Due to its susceptibility to climate fluctuations, vegetation phenology serves as a sensitive indicator for unraveling shifts in terrestrial ecosystems and the environment. Vegetation phenology, on one hand, responds to climate change as an impacted indicator; on the other hand, it acts as an influential factor that reciprocally impacts climate change, notably by regulating hydrothermal carbon exchanges within the soil–plant–atmosphere system, thereby amplifying climate change. Therefore, it acts as a double-edged sword, influencing and being influenced by climate change, with far-reaching ecological and environmental consequences. As a paramount indicator within vegetation phenology, the start date of the vegetation growing season (SOS) exhibits an exceptionally rapid response to climate change [5,6,7,8]; thus, monitoring the dynamics of the SOS, exploring the factors influencing the SOS, and understanding the relationship between the SOS and climatic factors are important for enriching our knowledge on the influence of climate change on terrestrial ecosystems.

Traditional phenological monitoring usually relies on ground observation. Although this traditional method could accurately record the timing of phenological events for a specific site and species [9], it is constrained by several limitations. Firstly, it is labor- and time-intensive, necessitating both fixed-point and species-specific observations. Secondly, the coverage of ground observation sites is often restricted [10], particularly in challenging environments such as alpine tundra and arid deserts [11]. Finally, the lack of a standardized protocol for ground observations leads to variations in phenological definitions among observers, thus impeding data exchange and harmonization. Therefore, it is difficult to monitor vegetation phenology over large areas and long time series [12]. Over the past few decades, the rapid evolution of satellite technology has provided an efficient platform for phenological monitoring on regional and global scales [1]. Remote sensing data have many advantages, such as global coverage and temporal continuity. These characteristics of remote sensing data compensate for the shortcomings of traditional vegetation phenology observation methods, rendering them extensively applied in regional and worldwide vegetation phenology research [3,4,12]. The Normalized Difference Vegetation Index (NDVI) has been extensively utilized in quantitative research of vegetation phenology due to its robustness and simplicity [13,14,15]. According to the majority of previous vegetation phenology studies based on satellite data, the SOS advanced owing to global warming [16]. However, the speed of advancement has varied in different regions, and even within the same region, the rate of advancement may vary considerably among different vegetation types. Mastering the dynamics of vegetation phenology can provide fundamental support for regional vegetation restoration, ecosystem protection, and the development of regional animal husbandry [17].

It has been confirmed that climate change can directly affect vegetation phenology [3,18,19]. Temperature and precipitation are widely acknowledged as primary natural factors influencing vegetation phenology [20,21,22]. Many studies have shown that the SOS in the Northern Hemisphere advanced with climate warming, including in Xinjiang [14,23], the Mongolian Plateau [24,25], and other regions. Moreover, recent studies have found asymmetric effects of daytime and nighttime warming on the SOS [26,27]. Notably, these asymmetric effects are complex. It remains uncertain whether maximum temperature (T_max_) or minimum temperature (T_min_) contributes more to SOS changes, and this may change based on the study area, vegetation type, and season. It was revealed that winter T_min_ had a stronger impact on the SOS compared to T_max_ on the Tibetan Plateau [20]. In contrast, in the arid and semi-arid temperate grasslands of China, it was indicated that there was a stronger relationship between SOS and winter T_max_ rather than T_min_ [27]. However, the situation was reversed in the spring, meaning that SOS was more closely related to T_min_. Sufficient water supply is a crucial factor for plant growth. Precipitation is a primary source of water for vegetation growth, thus having a significant impact on SOS [28]. If the temperature rises in the presence of water scarcity, SOS will not occur earlier. In fact, this situation might even lead to a further delay in the SOS, as rising temperatures exacerbate water evaporation. Overall, previous studies on vegetation phenology have recognized the important effects of air temperature and precipitation on vegetation. However, few studies have explored the importance of Land Surface Temperature (LST) in SOS changes. The importance of LST to vegetation phenology has been confirmed over the Great Lakes Region of Central Asia [29]. Therefore, delving into the effect of LST on the SOS contributes to a comprehensive understanding of the response mechanisms of SOS to climate change. While numerous scholars have paid attention to the impact of climate change on vegetation phenology, only a few have thoroughly investigated the predominant influencer among the various triggers of phenology change, including air temperature, LST, and precipitation.

The Tibetan Plateau (TP) is recognized as the world’s third pole. It supports an array of climate and ecosystem types ranging from tropical to cold and humid to arid due to its unique natural conditions [30,31,32]. As a sensitive and ecologically fragile zone of global change, the TP is considered an ideal place to study the response mechanisms of terrestrial ecosystems to global change [33]. Although some scholars have investigated the relationships between the SOS and climate change, there has been little progress towards determining the predominant driver of the SOS in the TP. Therefore, the principal aims of this study were as follows: (1) to quantitatively assess the spatial and temporal features of the current SOS variation and the consistency of the future SOS; (2) to investigate the response of the SOS to climate change over the entire area and among the vegetation types and to determine the predominant climatic factors (air temperature, LST or precipitation) affecting the SOS. This study contributes to the understanding of the mechanism of SOS variation in the TP.

## 2. Results

### 2.1. Spatial Patterns of the SOS

Significant spatial heterogeneity was observed in the vegetation phenology across the TP (Figure 1a). Overall, during the period of 2001–2020, the average value of SOS was 128 DOY (day of a year). In the past two decades, SOS in nearly 85.76% of the study area occurred from 100 DOY to 170 DOY (April to June). A relatively early SOS started in the east, while a later SOS occurred in the west. The variations in SOS showed diverse phenological patterns among meadows, steppes, shrubs, and forests (Figure 1b). Among all vegetation types, the SOS of meadows occurred earliest, predominantly varying from 83 through 150 DOY, with a mean SOS of 124 DOY; the SOS of shrubs started slightly later than that of meadows, mainly in the range of 92 to 150 DOY, with a mean SOS of 126 DOY; the SOS of forests was slightly later than that of shrubs, primarily varying between 90 and 170 DOY, with a mean SOS of 127 DOY; the SOS of steppes occurred the latest, covering a range of 83 to 180 DOY, with a mean SOS of 133 DOY.

### 2.2. Spatiotemporal Variation of the SOS

Overall, the SOS exhibited a slightly earlier trend at a rate of 0.261 days/year across the entire study region during 2001–2020. From a pixel-based perspective, there were two distinct trends in SOS changes: advance or delay. The SOS in approximately 63.59% of the study area demonstrated an advancing trend, of which 5.20% was significant (*p* < 0.05), mainly advancing by 0 to 1 day. Areas with an advancing SOS trend were predominantly located in Qinghai Province and its surrounding area (Figure 2). Only 36.41% of the study area showed a delayed trend in SOS, of which 1.03% was significant (*p* < 0.05), normally delaying by 0 to 3 days. Specifically, the SOS with a delayed trend was primarily found in the eastern region of the TP. Similar to the SOS observed in the entire study area, different vegetation types also exhibited an advancing trend in their SOS. Specifically, over the past 20 years, meadows displayed the fastest average change rate in SOS, advancing at a rate of 0.35 days/year. Steppes followed closely, with an advancement rate of 0.24 days/year. Shrubs had a marginally slower advancement rate than steppes, at 0.23 days/year. Finally, forests presented the most gradual advancement in SOS, at a modest rate of 0.17 days/year.

### 2.3. Consistency of SOS Trends

By using the SOS data for the years 2001 to 2020, the Hurst exponent was calculated to identify the spatial pattern of vegetation SOS consistency in the study region (Figure 3a). The findings showed that the areas with a Hurst exponent exceeding 0.5 were relatively limited and dispersed, mainly in the central and eastern parts, covering about 20% of the area. It is expected that these areas will continue their current trends into the future. In contrast, approximately 80% of the remaining region generally showed inconsistent characteristics (H < 0.5). Based on the Theil–Sen trend, the regions with consistent delay and inconsistent advance in SOS accounted for 7.72% and 45.73%, respectively, and were broadly distributed in the central and eastern parts of the TP (Figure 3b). Conversely, areas exhibiting consistent advance and inconsistent delay constituted 13.06% and 33.49%, respectively, and were mainly concentrated in the southwestern and central TP.

### 2.4. Relationships between the SOS and Climatic Factors

The PLSR model was utilized to investigate the influence of climatic factors on vegetation phenology changes. For the SOS, the PLSR model incorporated the following variables: T_max_, T_min_, daytime LST, nighttime LST, and precipitation in the previous year’s winter and the current year’s spring. The results showed that the VIP values for daytime LST, precipitation, and T_max_ in the previous winter, along with springtime T_max_, T_min_, and daytime LST, were all greater than 0.8 (Figure 4). The climatic factors with the greatest explanatory power for SOS changes, ranked from highest to lowest, were the previous winter’s T_max_ (VIP = 1.611), spring T_max_ (1.358), spring daytime LST (1.151), and winter precipitation (1.127). Except for T_max_ and daytime LST, positive model coefficients (MC) for the remaining variables in the previous winter indicated that increases in T_min_, nighttime LST, and precipitation all contributed to the postponement of the SOS. However, warming in T_max_ and daytime LST would advance the SOS date. Notably, there were asymmetric effects of daytime and nighttime warming (air temperature and LST) on SOS in the previous winter. The SOS was more strongly associated with T_max_ (with a higher VIP value and absolute MC) than with T_min_ in the previous winter, while the SOS exhibited a stronger correlation with nighttime LST than with daytime LST in the previous winter. With the exception of precipitation and nighttime LST, negative MC values for the other variables in the current spring indicated that increases in T_max_, T_min_, and daytime LST all contributed to an advance in the SOS. By contrast, an increase in spring nighttime LST and spring precipitation would delay the SOS date. Consistent with the previous winter’s LST, spring daytime LST and nighttime LST also exhibited asymmetric effects on SOS. The SOS was more strongly associated with T_min_ than with T_max_ in spring.

Across all four vegetation types, the SOS of each type showed diverse responses to climate variables. For forest (Figure 5a), negative MC values between SOS and precipitation indicated that increases in precipitation may have an advancing effect on SOS. Regarding the absolute MC and VIP values, the relationship between the SOS and winter precipitation displayed a much stronger association compared to that with spring precipitation. Based on the response of SOS to air temperature, except for winter T_max_, other air temperature variables exhibited negative MC values. According to the VIP values, winter air temperature exhibited relatively low VIP values (VIP < 0.8), suggesting no significant effects of winter T_max_ and T_min_ on the SOS. Conversely, higher VIP values (VIP > 0.8) of spring air temperature indicated significant impacts of spring T_max_ and T_min_ on SOS. The greater absolute MC and VIP values for spring LST compared to winter LST indicated a stronger linkage between spring LST and SOS.

For meadow (Figure 5b), despite the negative relationships observed between SOS and seasonal precipitation, a comprehensive analysis of absolute MC and VIP values revealed that the impact of spring precipitation on SOS was significantly greater than that of winter precipitation. Warmer air temperatures in the winter and spring could advance the SOS date. Notably, the SOS was asymmetrically affected by daytime and nighttime warming in spring, with T_min_ having a more pronounced influence. Regarding LST, it can be observed that the SOS demonstrated a stronger response to nighttime LST compared to daytime LST.

For shrub (Figure 5c), increasing precipitation in spring greatly contributed to an advance in the SOS. Regarding the effects of air temperatures on SOS, except for winter T_max_, all other air temperature variables advanced the SOS. Spring air temperatures had significant negative impacts. Similar to meadow, there also existed significant asymmetric impacts of daytime and nighttime warming in the spring on the SOS, with the SOS more strongly influenced by T_min_. As for LST, winter LST warming was found to potentially delay the SOS. By contrast, spring LST warming advanced the SOS. There were significantly asymmetric effects of daytime and nighttime LST in spring on SOS, with the SOS more strongly influenced by nighttime LST.

For steppe (Figure 5d), both winter precipitation and winter T_max_ were strongly associated with the SOS. Specifically, an increase in winter precipitation corresponded to a delay in the SOS. Conversely, winter T_max_ warming corresponded to an advance in the SOS. Regarding the SOS’s response to LST, the MC and VIP results suggested that spring nighttime LST exerted the most pronounced and negative influence on the SOS.

## 3. Discussion

### 3.1. Changes in the SOS on the TP

In the study, phenology was derived from a long-term time series of NDVI data by using the dynamic threshold method. The range of the SOS in the TP (100 DOY–170 DOY) was slightly inconsistent with the research of Wang et al. [34] (110 DOY–170 DOY; 1986–2015) and Wang et al. [35] (120 DOY–170 DOY; 2000–2015). This study found an expanded range of early SOS occurrence compared to the two previous studies. The slight discrepancy may be attributed to different NDVI datasets and study durations. Additionally, this may be linked to one of the findings of this study, which was the overall trend of advancing SOS. Although the studies mentioned above concluded the SOS calculation period in 2015, our study extended through 2020. This implied that SOS was still progressing beyond 2015, potentially broadening the scope of earlier SOS. Spatially, the SOS showed a delay from east to west. Influenced by the high altitude and arid water-thermal conditions, the SOS in the TP is usually much later than that in other low-altitude regions with abundant precipitation and favorable thermal conditions [29,36].

In response to climate change, vegetation phenology undergoes alterations. Nevertheless, distinct phenological stages display variations in both their direction and rate of change. Over the past 20 years, we found that the SOS exhibited an advancing trend across the entire region, which was basically in line with other studies in the Northern Hemisphere [3,22,27]. The average annual temperature in the TP has increased twice as much as the global average over the past five decades [37]. Due to the significant impacts of climate change, the SOS in the TP advanced at a rate of 0.261 days/year, surpassing that observed in Xinjiang (−0.19 days/year) and China’s temperate grasslands (−0.184 days/year). Although all four vegetation types exhibited an advancing trend in SOS, their rates of change varied widely due to differences in the response of each type to climate change. Among these, meadows showed the steepest advancement trend, with a rate of 0.35 days/year, aligning with the finding of Xu et al. [32]. This sensitivity may stem from the fragile and vulnerable ecological system of meadows, making them particularly susceptible to climate change. Since the mean value of H in the TP was 0.41 (H < 0.5), the future trend of the SOS changes may be opposite to the current trend, with an overall delayed trend. Specifically, about 80% of the region may show a delayed trend, whereas 20% of the region shows an advancing trend. This potential reversal could be linked to effective efforts to mitigate climate change by reducing the release of greenhouse gases through human intervention.

### 3.2. Response of the SOS to Climate Change on the TP

The response of phenology to climate change is intricately influenced by the combined effects of diverse climatic factors. Presently, a multitude of studies have substantiated temperature and precipitation as pivotal driving forces behind phenological shifts [13,38]. Temperature induces phenological changes by directly or indirectly impacting plant physiological processes [39]. Under the influence of temperature, a range of plant physiological activities, including photosynthesis [40], respiration [21], and water uptake, undergo adaptations. Generally, heightened temperatures engender heightened photosynthetic and metabolic activities, thereby promoting plant growth. Meanwhile, precipitation primarily influences vegetation phenology by supplying essential moisture to plants. The provision of sufficient water is of paramount importance for the growth and development of plants, as it not only supports the maturation of plant root systems but also facilitates nutrient assimilation and bolsters photosynthetic processes [41]. Consequently, these intricate processes collectively orchestrate the phenological trajectory of plants. In addition, water may regulate the controlling intensity of temperature on vegetation growth [28]. Over the past few years, an increasing number of studies have embarked on exploring the role of LST in vegetation phenological changes [42,43]. Similar to air temperature, LST regulates vegetation phenology by influencing plant physiological processes. Nonetheless, in contrast to air temperature, it also exerts control over vegetation phenology through its impact on soil temperature. Soil temperature is an intimate factor in the development of plant root systems and their capacity for water absorption, rendering it an indispensable determinant of vegetation growth [44]. Although vegetation phenology is subject to the combination of various contributing factors, one particular driver stands out as dominant. However, the leading factor is not fixed and varies according to the study area and the specific phenological types. This study demonstrated that in the TP’s vegetated areas, concerning the impact of climatic factors on phenology during winter and spring, the preeminent role is assumed by winter T_max_ (as evidenced by the highest VIP value). Additionally, a negative correlation exists between SOS and winter T_max_, implying that elevated winter T_max_ results in an earlier initiation of the SOS. There might be two contributing factors to this phenomenon. On one hand, plants require a certain level of accumulated warmth to begin their spring growth [26]. Warmer winter T_max_ could speed up the accumulation of required warmth, causing an earlier SOS. On the other hand, high-altitude regions often experience harsh winter conditions with freezing and thawing cycles. If winter T_max_ rises, snow and ice duration could be shortened, causing soil to thaw earlier. This allows plants to access water and nutrients sooner, advancing SOS. Furthermore, winter T_max_, the impact of spring T_max_ on the SOS should not be overlooked. A negative correlation exists between spring T_max_ and the SOS, signifying that elevated spring T_max_ results in an earlier SOS. Furthermore, irrespective of air temperature or LST, our study identified a significant diurnal asymmetry in the influence of winter daytime and nighttime warming on the SOS: the SOS was more strongly associated with T_max_ rather than T_min_ in the winter. Similar results were found in arid and semi-arid temperate grasslands in China [27]. In the TP, there is a significant temperature difference between day and night, with T_min_ much lower than T_max_. Before the onset of vegetation growth, T_min_ had a higher likelihood of falling below the threshold temperature compared to T_max_, hence contributing less to meeting the thermal requirement for vegetation growth [26,45]. Therefore, warming winter T_max_ is more effective than warming T_min_ in meeting the thermal requirements necessary to trigger the SOS.

For the four distinct vegetation types, the dominant factor responsible for SOS variations exhibits variability. For forest, among all factors, spring nighttime LST exerted the most substantial influence on SOS, presenting a negative effect. This phenomenon could be attributed to two main reasons. On one hand, spring nighttime LST warming could accelerate soil thawing, providing more nutrients and water for vegetation growth; on the other hand, nighttime LST can reach an extremely low level, even resulting in damage to vegetation. Therefore, spring nighttime LST warming may protect vegetation from the threat of cold damage in early spring, thus making vegetation more likely to commence growth earlier [46]. According to the VIP results, the impact of winter precipitation on SOS was second only to that of spring nighttime LST. Winter precipitation also had a negative effect on the SOS. For meadow and shrub, spring air temperatures were the dominant factors affecting their SOS changes. Additionally, there was a negative correlation between them. Moreover, the effects of T_max_ and T_min_ on the SOS were asymmetric in the spring: the SOS was more strongly associated with T_min_ than with T_max_. The greater effects of rising T_min_ were likely related to the premature termination of plant dormancy. Dormancy is a survival mechanism adopted by plants to endure cold conditions [31,47]. Therefore, plants commonly enter a dormant state in regions with frigid winter temperatures and diminished sunlight. Consequently, an increase in spring T_min_ can be interpreted as a signal for the safe termination of dormancy, thereby initiating the growth phase. Furthermore, higher T_min_ can provide more available soil water from ice and snow. This increased availability of soil water allows vegetation roots to absorb enough water to prepare for early leaf spread [48]. Conversely, the limited influence of higher T_max_ on the SOS may be due to the decrease in water availability [20]. Although an increase in spring T_max_ creates favorable thermal conditions for vegetation growth, it concurrently enhances water evaporation, leading to faster water loss. These dual effects tend to partially counterbalance each other, resulting in a relatively weaker impact on SOS compared to spring T_min_. Differing from other vegetation types, our study revealed that, for steppe vegetation, winter precipitation was the decisive factor for SOS, not air temperature or LST, which indicated that water availability may play the most crucial role in controlling the occurrence of vegetation growth. Winter precipitation was positively correlated with the SOS, similar to that observed in the Xinjiang region [14]. In the TP, winter precipitation predominantly takes the form of snowfall, resulting in a lower temperature. Additionally, the snow cover diminishes incoming solar radiation, thereby affecting vegetation growth.

## 4. Materials and Methods

### 4.1. Study Area

Nestled in the southwestern part of China (Figure 6a), the Tibetan Plateau (TP) stands as the loftiest plateau on Earth, boasting an average elevation of approximately 5000 m. The TP (26°10′ N–39°47′ N, 74°19′ E–104°47′ E) spans an extensive latitude and longitude range of approximately 14° in the north–south direction and up to 30° in the east–west direction. The TP covers an area of 2,572,400 km^2^, accounting for 26.8% of China’s land area. It incorporates not only Tibet and Qinghai but also the southern part of Xinjiang, the western section of Sichuan, and the northwest corner of Yunnan. The internal terrain of the plateau is intricate and varied, featuring a significant elevation range (82–8405 m) (Figure 6b). Generally sloping from northwest to southeast, it demonstrates distinct vertical zones with notable differences in phenology at varying elevations. This area exhibits unique climatic characteristics resulting from its intricate topography and high elevation, including intense solar radiation, frigid temperatures, and low and irregular precipitation patterns.

The unique ecological surroundings in the TP nurture its diverse species and complex ecosystems. The vegetation types in the region are multiple (Figure 6c), with mosaic distribution and certain regularity. From northwest to southeast, there are many vegetation types such as bare desert, arable land, alpine steppe, alpine meadow, mountain shrub, and mountain forest. Among these vegetation types, alpine steppe (29.99%), alpine meadow (22.33%), shrub (10.27%) and forest (6.22%) make up most of the land cover on the TP. Thus, we took these four vegetation types as research objects and discussed the connection between phenological events of four vegetation types and various climatic factors.

### 4.2. Data Sources

#### 4.2.1. NDVI Data

NDVI is a reliable indicator for assessing vegetation growth, which is widely utilized for measuring growth status and processes in plants [49]. In this study, the NDVI data were acquired from the MOD13A2V006 dataset published by NASA (https://ladsweb.nascom.nasa.gov/search/ (accessed on 8 March 2023)). The dataset possessed a spatial resolution of 1 km and a temporal resolution of 16 days, covering the study duration from 2001 to 2020. The original dataset has been preprocessed by geometric calibration, radiometric calibration, and atmospheric calibration [22]. However, noise still remained in some pixels. Hence, in this paper, Savitzky–Golay [50,51] filtering was used for smoothing to reduce the noise in the data.

#### 4.2.2. Climate Data

In this study, daily climate data including daily precipitation, daily maximum temperature (T_max_), and daily minimum temperature (T_min_) from 2001 to 2020 were downloaded from the Chinese National Meteorological Center (NMC) (http://data.cma.cn./ (accessed on 8 March 2023)). Considering the continuity and integrity of the climate data series, we selected 117 meteorological stations with continuous data records in the TP and its surrounding areas (Figure 1b). The specialized meteorological software ANUSPLIN was utilized to achieve the spatial extension of meteorological data from point to surface. ANUSPLIN incorporates spline interpolation theory to simultaneously integrate linear submodels for multivariate covariates and spatially interpolate multiple surfaces. The LST data were obtained from NASA’s Terra satellite MOD11A2 Version 6 surface temperature product (https://espa.cr.usgs.gov/ (accessed on 8 March 2023)), with a spatial resolution of 1 km and a repeat cycle of 8 days. The Terra satellite transits twice a day at 10:30 and 22:30 local time, representing daytime and nighttime LST. We used the quality control files to eliminate the pixel data with poor quality, then utilized the domain mean to fill the null values. Finally, we synthesized the 8-day LST data into seasonal LST.

#### 4.2.3. Land Cover Data

The land cover data for the Tibetan Plateau were retrieved from the Science Data Bank, accessible at https://www.scidb.cn/doi/10.11922/sciencedb.398 (accessed on 8 March 2023). The data represent the spatial distribution of 12 vegetation types, such as alpine desert, alpine steppe, alpine meadow, permanent snow, bare land, and other types. Considering the coverage of natural vegetation areas, we extracted four vegetation types from the data for zonal study, including alpine steppe, alpine meadow, shrub, and forest.

### 4.3. Methods

#### 4.3.1. Phenological Extraction

At present, the most widely used vegetation phenology extraction methods mainly include the dynamic threshold method [30,52], polyfit-Maximum method [25,53], and Delayed Moving Average method [54,55]. While the methods mentioned above are widely used, their selection should be determined based on the characteristics of the specific region. The polyfit-Maximum method requires a higher level of stability in vegetation growth conditions, making it unsuitable for regions where external factors significantly impact vegetation growth. The Delayed Moving Average method requires a higher level of vegetation cover and is less suitable for regions with significant fluctuations in time-series curves or poor vegetation cover. In contrast, the dynamic threshold method has fewer strict requirements and offers greater flexibility and adaptability. Therefore, in this study, the dynamic threshold method was chosen to extract the SOS. In this study, based on TIMESAT 3.3, we first used Savitzky–Golay filter method to smooth the NDVI data with the aim of reducing noise and reconstructing the time series and then combined it with the dynamic threshold method to extract the SOS during the study period. For every pixel, the relative change in NDVI was determined through the following equation:$$NDVI_{ratio}=\frac{NDVI-NDVI_{min}}{NDVI_{max}-NDVI_{min}}$$
where *NDVI_ratio_* is the *NDVI* threshold; *NDVI* is the normalized vegetation index after smoothing; and *NDVI_max_* and *NDVI_min_* indicate the peak and trough of the yearly NDVI fluctuations, respectively. Referring to the existing literature data, the extraction threshold of the SOS was generally set to 0.2 [56,57,58,59]. In this paper, the extraction threshold was also set to 0.2, which could accurately depict the actual status of vegetation growth in the TP.

#### 4.3.2. Theil–Sen Trend

To monitor the interannual variation trends of vegetation phenology in the TP during 2001–2020, we applied the Theil–Sen trend to examine the trends of vegetation phenology at the pixel level. The Theil–Sen trend is a reliable nonparametric statistical method for calculating trends, which has no requirements on the distribution of data and can reduce the interference of sample outliers [19]. For long time series data, we firstly calculated the slope between any two points and then sorted the slopes in ascending order. Finally, the intermediate slope was selected as the trend of the whole sequence. The formula for calculation is as follows:$$\beta =Median\left(\frac{x_{j}-x_{i}}{j-i}\right),n>j>i\geq 1$$
where *β* represents the phenological trend slope; n is the length of the time series; *x_i_* and *x_j_* represent the values of year *i* and year *j*, respectively; and *Median* () represents the median function, that is, the intermediate value of the slope series.

#### 4.3.3. Mann–Kendall Test

The Mann–Kendall test is a classical nonparametric trend test method to evaluate the significance of Sen’s slope. During the test, the Mann–Kendall test does not require the data to follow specific rules and is not disturbed by a small number of outliers. It is suitable for trend analysis of non-normal data such as phenology and precipitation [60]. The specific calculation formula is as follows:$$Z=\left\{\begin{array}{c} \frac{S-1}{\sqrt{Var\left(S\right)}},\;S>0\\ 0,\;S=0\\ \frac{S+1}{\sqrt{Var\left(S\right)}},\;S<0 \end{array}\right.$$

The formulas of *S* and *Var* (*S*) is as follows:
$$S=\overset{n-1}{\underset{i=1}{\sum }}\overset{n}{\underset{j=i+1}{\sum }}sgn\left(x_{j}-x_{i}\right)\;sgn=\left\{\begin{array}{c} +1\;x_{j}>x_{i}\\ 0\;x_{j}=x_{i}\\ -1\;x_{j}<x_{i} \end{array}\right.\;Var\left(S\right)=\frac{n\left(n-1\right)\left(2n+5\right)}{18}$$
where *x_i_* and *x_j_* are the *i*th and *j*th data values in the time series, respectively; *sgn* is a function symbol; and *n* is the year of the time series. In the test, we generally determine whether the result is significant by giving a confidence level value α. If |*Z*| ≧ *Z*_1 − *a*_ / 2, the trend of the time series data is significant. In general, when |*Z|* ≧ 1.65, 1.96, and 2.58, the trend of change is examined by the significance test with confidence levels of 90%, 95%, and 99%, respectively [61].

#### 4.3.4. R/S Analysis

Self-similarity and long-term dependence are prevalent phenomena in nature. The Hurst exponent is an effective indicator for characterizing these phenomena. In this study, the rescaled-range analysis method was applied to calculate the Hurst exponent. The formula is as follows. Given the time series {*SOS* (*t*)}, *t* = 1, 2, ..., *n*. For any positive integer *m* ≥ 1, define the mean, cumulative deviation, extreme deviation, and standard deviation in turn.

(1) Define the mean sequence:
$$\bar{SOS\left(m\right)}=\frac{1}{m}\overset{m}{\underset{t=1}{\sum }}SOS\left(m\right)\;m=1,2,\cdots ,n$$

(2) Calculate the cumulative deviation:
$$X\left(t,m\right)=\overset{m}{\underset{t=1}{\sum }}\left(\mathrm{SOS}\left(m\right)-\bar{\mathrm{SOS}\left(m\right)}\right)\;1\leq t\leq m$$

(3) Define the extreme deviation:
$$R\left(m\right)=\mathrm{maxSOS}\left(t,m\right)-\mathrm{minSOS}\left(t,m\right)\;m=1,2,\cdots ,n$$

(4) Calculate the standard deviation:
$$S\left(m\right)={\left[\frac{1}{m}\overset{m}{\underset{t=1}{\sum }}{\left(\mathrm{SOS}\left(t\right)-\mathrm{SOS}\left(m\right)\right)}^{2}\right]}^{\frac{1}{2}}$$

Define *R*/*S* = *R*(*m*)/*S*(*m*). If *R*/*S*∝*m*^H^, then the Hurst (H) phenomenon exists in the time series. The Hurst exponent (H) lies between 0 and 1. (1) When 0.5 < H < 1, the future trend may align with the past trend, with higher consistency as H approaches 1. (2) When 0 < H < 0.5, the future trend displays inconsistency and will reverse in the future. (3) When H = 0.5, the future trend is unpredictable.

#### 4.3.5. Partial Least Squares Regression (PLSR)

In this study, partial least squares regression was applied to explore the relationship between spring phenology and climatic factors. The PLSR method combines the benefits of canonical correlation analysis, principal component analysis, and multiple linear regression analysis to avoid multicollinearity among independent variables [62]. In addition, this method provides a noteworthy advantage by allowing for the utilization of more independent variables than the quantity of available samples. Two indicators of each variable can be obtained by PLSR calculation. One is the standard regression coefficient, and the other is the variable importance projection (VIP). The VIP value usually reflects the contribution of each independent variable to the change in dependent variables [63]. The expression of the VIP value is as follows:$$VIP_{j}=\sqrt{p\sum_{h=1}^{m}\sum_{i}R^{2}\left(y_{i},t_{h}\right)w_{hj}^{2}/\sum_{h=1}^{m}\sum_{i}R^{2}\left(y_{i},t_{h}\right)}$$
where *p* is the number of independent variables; *m* is the number of components extracted from the independent variables; *i* represents the *i*th dependent variable; *t_h_* represents the *h*th component of the independent variables; *R*^2^(*y_i_*,*t_h_*) denotes the square of the correlation coefficient between *y_i_* and *t_h_*; and $w_{hj}^{2}$ represents the contribution weight of the independent variable *x_j_* to the construction of the *t_h_* component. According to previous studies, when the VIP values of climatic factors exceed 0.8, it shows that the climatic factors have certain explanatory significance for phenological changes; when the VIP value is greater than or equal to 1, it indicates that the variables have obvious explanatory significance [64].

## 5. Conclusions

In this study, we first analyzed the spatial and temporal variation of spring phenology of vegetation over the TP during 2001–2020. The results showed that, for nearly the whole study area, the SOS concentrated on 100 DOY–170 DOY, with a delay from east to west. Although the SOS across the entire region exhibited an advancing trend at a rate of 0.261 days/year, there were notable differences in the advancement trends of SOS among different vegetation types. In contrast to the past trend in the SOS changes, the future SOS may be delayed. Then, we explored the response of SOS to climate changes by using PLSR and determined the dominant factor affecting the SOS. Winter T_max_ played the dominant role in the temporal shifting of spring phenology across the TP, and its effect on SOS was negative, meaning that an increase in winter T_max_ led to an earlier SOS. For forest, spring nighttime LST made the biggest contribution to the shifting of the SOS, with higher spring nighttime LST contributing to an advance in the SOS. For meadow and shrub, spring air temperatures were the dominant factors affecting their SOS changes and were negatively correlated. In addition, the effects of T_max_ and T_min_ on the SOS were asymmetric in the spring: the SOS was more strongly related to T_min_ than to T_max_. However, for steppe, our study revealed that the decisive factor influencing SOS was winter precipitation, rather than air temperature and LST.

Our investigation exclusively addressed the influence of climatic factors on vegetation phenology. However, non-climatic factors have attracted increasing attention in recent years. Human activities, such as grazing, emissions, and land use, have been documented to influence vegetation phenology [12,65]. Consequently, future research endeavors should aim to comprehensively explore the combined effects of climatic and non-climatic factors on vegetation phenology.

## Figures and Tables

**Figure 1 plants-13-01065-f001:**
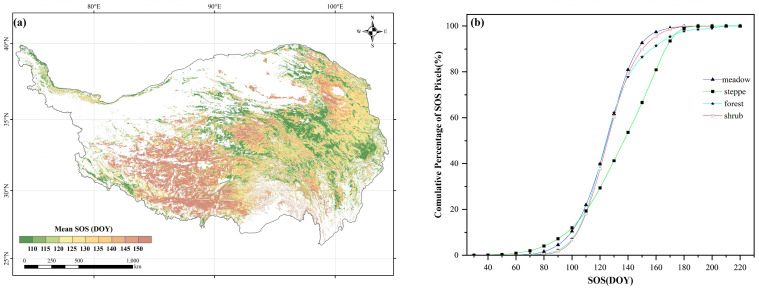
(**a**) The spatial distribution of the annual mean SOS; (**b**) the cumulative percentage of the SOS pixels for the four vegetation types.

**Figure 2 plants-13-01065-f002:**
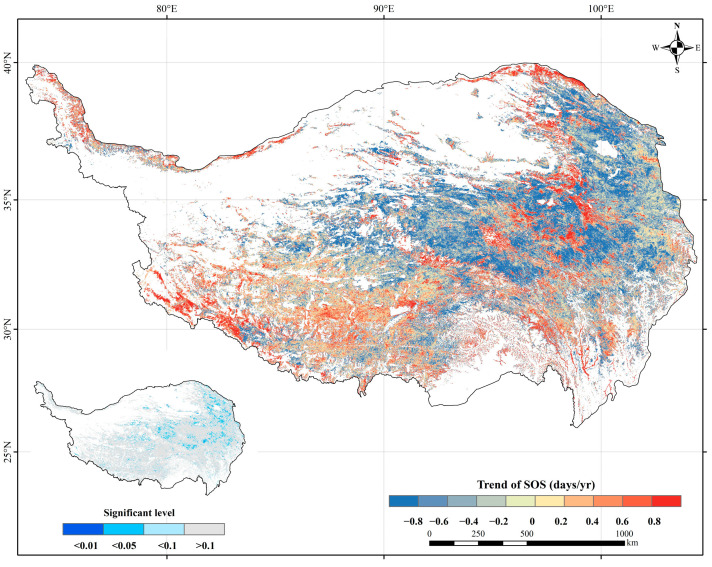
Spatial distribution of the trend in the SOS for the period 2001–2020. The picture in the lower left corner shows the significance level of the trend.

**Figure 3 plants-13-01065-f003:**
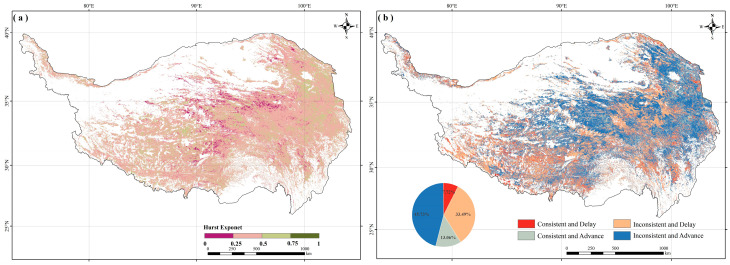
The spatial distribution of (**a**) the H values of the SOS and (**b**) the future trend of the SOS.

**Figure 4 plants-13-01065-f004:**
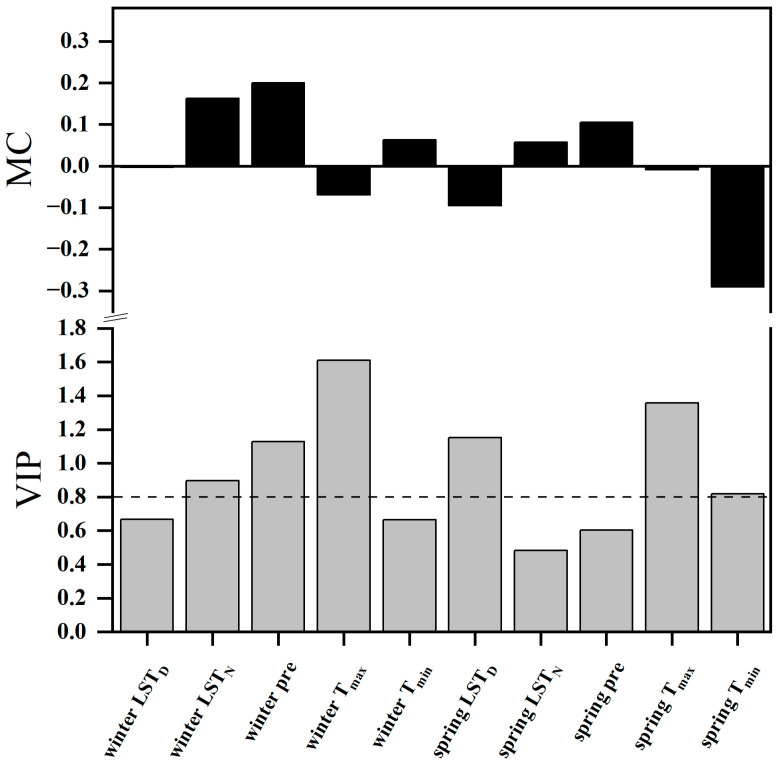
Responses of the SOS in the TP to climatic factors (daytime LST, nighttime LST, pre, T_max_, T_min_) in the previous winter and the current spring according to partial least squares (PLS) regression (2001–2020). The top graph represents model coefficients (MC), and the bottom graph represents the variable importance plots (VIP) values. In the picture, pre stands for precipitation, LST_D_ for daytime LST, and LST_N_ for nighttime LST.

**Figure 5 plants-13-01065-f005:**
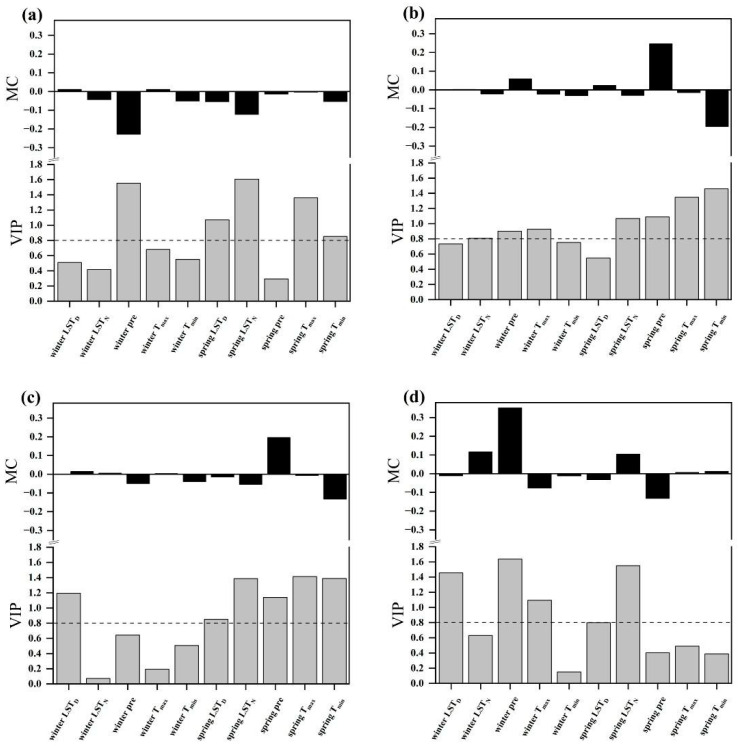
Responses of the SOS for four vegetation types to climatic factors (daytime LST, nighttime LST, precipitation, T_max_, T_min_) in the previous winter and the current spring according to partial least-squares (PLS) regression (2001–2020). (**a**) forest; (**b**) meadow; (**c**) shrub; (**d**) steppe.

**Figure 6 plants-13-01065-f006:**
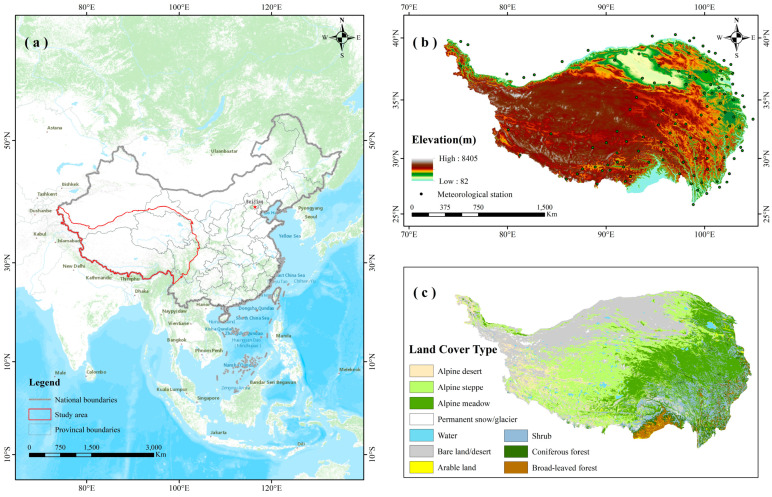
(**a**) The location of the TP within China; (**b**) DEM and the spatial distribution of the meteorological stations in the TP; (**c**) land cover types.

## Data Availability

The data that support the findings of this study are openly available in the NASA MODIS Portal (https://modis.gsfc.nasa.gov/ (accessed on 8 March 2023)), daily climate data (http://data.cma.cn./ (accessed on 8 March 2023)), the land cover data (https://www.scidb.cn/ (accessed on 8 March 2023)).

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
