# Peer review of "Variation of the Start Date of the Vegetation Growing Season (SOS) and Its Climatic Drivers in the Tibetan Plateau"

_plants, 2024, doi:10.3390/plants13081065_

Round 1

Reviewer 1 Report

Comments and Suggestions for Authors

Dear Authors!

The manuscript is not prepared according to the Plants' Journal Instructions (order of the sections and citation style in the text).

English language editing is needed. Some specific comments/recommendation I have provided in Specific comments section.

Title is too broad, given the fact that You have analysed just one aspect of the phenology (even only the spring aspect of it, since it could include flowering, seedling emergence etc.). Hence, it should be rewritten to better reflect the content of the manuscript.

Too extensive usage of the abbreviations in the text overall (I really do not see a rationale to use „pre“ for „precipitation“ throughout the manuscript), without explaining their meaning on first mentioning (e.g. TB, SOS, LST). These are explained in the Abstract, but it should be explained in the main text of the manuscript as well.

There is lack of explanation in the Methods which methods were used to enable usage of climate data originating from the meteorological stations (point data) over entire area of interest.

Discussion at present, is too extensive given the proportion of its content dedicated to comparison with existing literature on the topic vs. commenting own results and some general facts about the climate dependent vegetation processes (proportion being in favour of the latter).

 Specific comments

 Line 61 – I am not sure to what are you referring here with this: „it is limited by natural environmental condition“? What is limited? Method of observing, start of the phenology phase, something else? Please rephrase this part.

Lines 75-77 – A bit of vague sentence (e.g. accurate investigations; hold profound significance). Please consider rephrasing with focusing on the importance of research on the “temporal dynamics of phenological phase”.

Line 110 – “evolution of terrestrial ecology” sounds strange. It is not clear to me whether you are writing about the discipline itself or about the dynamic/variation of processes taken place in the nature. Either way, I think you can omit this sentence.

Line 116 – consider replacing “in terms of the total area and vegetation types” with “over entire area and among the vegetation types” if that is what you wanted to communicate with readers.

Line 124 – range of geographical latitude is something below 14, not 15!

Line 128 – please use “range” here instead of “disparities”, and check the values, since in the text you are writing about 32 meters as minimum elevation, and on the Figure 1b according to the legend it is 82!

Figure 1b – something is wrong with the coordinate values on this figure, because there is a significant shift (disparity) between values on left vs. right border (latitude) and upper vs. lower border (longitude).

Line 156 – “to reduce the noise in the data” not “in the paper”!

Line 166 . delete “in the paper” at the end of this sentence, and start new one with capital letter.

Line 168 – Please, provide more details on the methods here, for this synthesis.

Line 171 – Please consider using here “represents” instead of “illustrate”

Line 186 – I presume here in the numerator it should be “NDVImin” instead of “NDVIratio”.

Line 188 – I am a bit confused why are you using “nadir” here? Please explain, or delete.

Line 267 – please consider using “Among” instead of “For” here.

Comments on the Quality of English Language

English language editing is needed. Some specific comments/recommendation I have provided in Specific comments section, but overall editing will be beneficial.

Reviewer 2 Report

Comments and Suggestions for Authors

In the manuscript “Variation of Vegetation Spring Phenology and Its Climatic Drivers in the Tibetan Plateau” the authors examine the responses of vegetation phenology to climate change through finding the dominant factors that induce changes in the start date of the vegetation growing season (SOS). They found differences in the advancement trends of SOS among four different vegetation types and they conclude that the leading factor causing changes in vegetation phenology was different in each one of the four vegetation types. Considering the importance of this kind of study to improve the knowledge about the impact of climate change on vegetation phenology, I believe that the manuscript is of potential interest to readers of “Plants” and falls within its scope.

Abstract: It briefly summarizes the major aspects of the study.
Introduction: It presents effectively the current knowledge on the subject, introducing well the role of the proposed study and the research gap.

Materials and Methods: They are clear and well detailed.
Results and Discussion: In general, they are clear and well-written.
Conclusion: It summarizes the main results observed in this study.

Minor Comments:

·        L. 86-87: Rewrite the sentence as follows “It remains uncertain whether Tmax or Tmin contributes more to SOS changes and this may change based on the study area, vegetation type and season”.

·        L. 93: Replace “key condition” by “crucial factor”.

·        The quality of the graphs in figure 1 and 2 should be improved.

·        L. 273: Replace “150 DOY” by “180 DOY

·        L. 491-497: This paragraph should be moved to the conclusions

Round 2

Reviewer 1 Report

Comments and Suggestions for Authors

Dear Authors!

My only comment on the revised manuscript is that you should solve the coordinates issue on Figure 6, as mentioned in my review on the original version of your manuscript.
